# Secondary Resistant Mutations to Small Molecule Inhibitors in Cancer Cells

**DOI:** 10.3390/cancers12040927

**Published:** 2020-04-09

**Authors:** Abdulaziz B. Hamid, Ruben C. Petreaca

**Affiliations:** Department of Molecular Genetics, The Ohio State University, Marion, OH 43302, USA; hamid.26@osu.edu

**Keywords:** resistant mutation, small molecule inhibitor, targeted therapy, cellular adaptation, combination therapy

## Abstract

Secondary resistant mutations in cancer cells arise in response to certain small molecule inhibitors. These mutations inevitably cause recurrence and often progression to a more aggressive form. Resistant mutations may manifest in various forms. For example, some mutations decrease or abrogate the affinity of the drug for the protein. Others restore the function of the enzyme even in the presence of the inhibitor. In some cases, resistance is acquired through activation of a parallel pathway which bypasses the function of the drug targeted pathway. The Catalogue of Somatic Mutations in Cancer (COSMIC) produced a compendium of resistant mutations to small molecule inhibitors reported in the literature. Here, we build on these data and provide a comprehensive review of resistant mutations in cancers. We also discuss mechanistic parallels of resistance.

## 1. Introduction

Paul Ehrlich who is famous for discovering a treatment for syphilis as well as making impactful contributions to the treatment of other bacterial infections was the first to use the word “chemotherapy” when referring to treatment of diseases with chemicals [1]. However, the use of drugs to treat cancer was pioneered by Sydney Farber during his work on childhood leukemia [2]. Because of Farber’s foundational contribution to the development of various anti-cancer drugs, today we think of chemotherapy as a cancer treatment.

Farber’s drug of choice was aminopterin, a competitive inhibitor of dihydrofolate reductase [3]. The drug blocks nucleotide synthesis which affects both DNA replication and transcription. It targeted the fast dividing leukemia cells in children leading to remissions in about 30% of the patients [4]. Farber and his colleagues were careful not to use the word “cured” when discussing these patients. The remission patients eventually developed resistance to the drug and ultimately succumbed to the disease. One answer was to administer a higher dose of the drug, but Farber was reluctant to do so because he was deeply affected by his patients’ suffering. Emil Freireich and Emil Frei working at the National Cancer Institute (NCI) proposed the use of combination chemotherapy. Using randomized studies, the NCI has been central in the discovery of a plethora of chemotherapeutic drugs, both synthetic and natural plant extracts [5]. In Siddhartha Mukherjee’s book, *The Emperor of All Maladies: A biography of Cancer*, the historical timeline of these events are outlined [6].

Advances in chemical synthesis combined with a deeper understanding of protein three dimensional structures have allowed identification of druggable targets [7,8]. Importantly, not all factors involved in a pathway can be druggable. For example, in a signal transduction pathway only certain key enzymes (e.g., tyrosine kinases) may be druggable [9]. Inhibiting these key enzymes is sufficient to inactivate the entire pathway. Most modern drug discovery focuses on identifying key targets that are druggable and synthesizing small molecule inhibitors [10,11,12]. This type of treatment with small molecule inhibitors is known as “targeted therapy.” We now use the word “chemotherapy” to refer only to general cell cycle inhibitors.

Cellular adaptation to targeted therapy has been an issue in cancer treatment from the very beginning and is the reason for Farber’s only temporary remissions. One form of adaptation is to compensate for the inhibition of one pathway by modulating other pathways and escape the drug’s inhibitory effects. Combination therapy is, therefore, more efficient in killing the cells because it targets several pathways at once. Another form is to acquire a mutation in the target gene making the protein resistant to the drug. To overcome the resistance, it is often possible to synthesize a second generation, or even third or higher generation drug that is potent even against the resistant mutations.

Unlike cancer driver mutations which generally appear before drug treatment [13,14,15], resistant mutations are secondary mutations that appear in response to adaptation to the drug [16]. Here we used the Catalogue of Somatic Mutations in Cancer (COSMIC) database (cancer.sanger.ac.uk) [17] to retrieve targeted therapy resistant mutations. We map these data and discuss the implication of these resistant mutations. Although some of these mutations have been reviewed before, this is a comprehensive review of all cancers with the aim of pointing out parallels between the various resistant mechanisms.

## 2. Data Acquisition and Manipulation

The “COSMIC resistance mutations” file was downloaded from the version 90 COSMIC database (https://cancer.sanger.ac.uk/cosmic/download). This file categorizes all the reported mutations and provides the PubMed ID for all queries. The information in this file was used to plot all the mutations in Figure 1, Figure 2, Figure 3, Figure 4, Figure 5 and Figure 6. Lollipop figures were made as previously described [18]. The Lollipops software uses Uniprot and Pfam to gather the protein information. Next to each mutation we indicated the number of times it appears in the file (e.g., X4). In our figures we restrict our diagrams of the targeted pathways to only those relevant to the drugs discussed. All references curated by COSMIC and listed in the file are provided in Appendix A. Some references provided in this file were also used in this report. Because the literature on this topic is vast, we could not include most of the references. Although we tried to cite primary research papers when referring to specific mutations, we often had to cite reviews when referring to broader aspects of the field. The goal of this review is to give an overview of the most reported resistance mutations.

## 3. Overview of Resistance Mutations

Using the COSMIC database, we retrieved resistance mutations to 37 therapies that target 23 proteins (Table 1). Some drugs are not specified (e.g., Tyrosine kinase inhibitors-NS). The most represented tissues were lung and hematopoietic and lymphoid while adenocarcinoma was the most likely tumor type to develop resistance. Resistance mutations arise most often in protein tyrosine kinase domains. The properties of different small molecule protein kinase inhibitors types have been discussed elsewhere [19,20,21]. We focus this review on secondary resistant mutations that arise in response to the therapy. Thus, we do not discuss all therapeutic drugs but only those that acquire secondary resistant mutations.

## 4. Resistant Mutations Appear in Several Cancers in Response to Small Molecule Inhibitors

### 4.1. Non-Small Cell Lung Cancer

#### 4.1.1. Classification

Lung cancer is classified into small cell lung cancer (SCLC) and non-small cell lung cancer (NSCLC). NSCLC is further sub-classified into adenocarcinoma, large cell carcinoma, and squamous cell carcinoma [23]. Bronchioalveolar (BAC) and acinar carcinoma are two forms of adenocarcinoma. Under the new classification of the Association for the Study of Lung Cancer/American Thoracic Society/European Respiratory Society (IASLC/ATS/ERS), the term “bronchioalveolar carcinoma (BAC)” has been proposed to be discontinued. BAC is currently re-classified as adenocarcinoma in situ, lepidic predominant adenocarcinoma, and minimally invasive adenocarcinoma [24]. However, since some COSMIC cases precede this classification, we will still use the term bronchioalveolar in this report. Squamous cell carcinoma and adenosquamous carcinoma are the two other forms of lung cancer [25] reported on COSMIC. “Non-small cell carcinoma” refers to non-classified NSCLC or unknown origin.

#### 4.1.2. Epidermal Growth Factor Receptor (EGFR)

Most NSCLC drug therapies target tyrosine kinases in several signal transduction pathways [26,27,28]. Several drugs have been approved for treatment of NSCLC [29,30]. We discuss the ones that affect the pathways presented in Figure 1A because resistance mutations have been reported. The epidermal growth factor receptor (EGFR or ErbB1) is a receptor tyrosine kinase from the ErbB family RTKs. This family also includes ERBB2 or HER2, ERBB3 or HER3, and ERBB4 or HER4 [31]. EGFR signals primarily through the PI3K-AKT-mTOR [32,33] and the RAS-RAF-MEK-ERK [34,35] pathways. Physiologically, the function of EGFR is to promote survival and proliferation. EGFR also activates CDC42 which is involved in cell migration [36,37]. JAK and STAT3 (Signal transducer and activator of transcription 3) are two other pathways activated by EGFR not shown in Figure 1A [31]. Most NSCLCs are characterized by EGFR overexpression [38,39] or activating mutations also known as “common mutations” (e.g., L858R, delL747-P753ins, G719S, several exon 19 deletions and exon 20 insertions) [40,41,42,43]. These mutations are found in the P-loop or activation loop of the kinase which are essential for function and are universally conserved in kinases [44,45,46]. Some of these mutations sensitize the kinase to first generation tyrosine inhibitors erlotinib and gefitinib (see next paragraph).

ERLOTINIB and GEFITINIB are first generation reversible tyrosine kinase inhibitors. Erlotinib (CP-358,774) is an ATP analog that inhibits EGFR tyrosine autophosphorylation [47]. Gefitinib (ZD1839) is an independently synthesized ATP-competitive inhibitor of EGFR [48,49]. When compared to the WT enzyme, the L858R mutation decreases the affinity of enzyme for ATP (higher K_m_) while simultaneously increasing its affinity for the drugs (lower K_i_) [50,51]. Continuous treatment with either erlotinib or gefitinib leads to the acquired EGFR T790M gatekeeper mutation (Figure 1E) [52,53]. The T790M mutation appears to compensate for the loss of ATP affinity caused by the L858R mutation. Double L858R T790M mutants show the same ATP affinity as WT cells [54,55,56]. The T790M resistant mutation occurs even when treating patients with EGFR activating mutations other than L858R. In at least one instance, erlotinib treatment of an EGFR (G719A L861Q) patient resulted in the T790M mutation. The patient also acquired the D1067V mutation in PIK3CB (Figure 1D) which leads to oncogenic transformation of this kinase [57]. The PIK3CB mutation which is in the kinase domain activates the PIK3 pathway in response to EGFR inhibition and leads to cellular transformation both in vitro and in vivo [57,58].

AFATINIB (BIBW2992) is a second-generation inhibitor that targets the entire ErbB receptor family [59,60]. Preclinical studies found that it inhibits the activity of WT enzyme as well as that of L858R single or L858R T790M double mutants [61]. Afatinib is an irreversible inhibitor because it forms a covalent interaction with C797 of EGFR and blocks the enzymatic activity of the kinase [62]. Remarkably, patients treated with afatinib that harbored EGFR common activating mutations also acquired the T790M resistant secondary mutation (Figure 1E, Appendix A) [63,64]. A clinical study also found that the T790M mutation was resistant to afatinib [65]. Some L858R afatinib treated patients also acquired a C797S mutation. Both the T790M and the C797S mutations rendered cells resistant to the drug [64]. Mapping the C797S mutation on the crystal structure of EGFR shows that it disrupts the ability of the drug to make a covalent bond with the kinase [64]. The L792F was another mutation identified by the same study to arise after treatment with afatinib. The L792F mutation causes steric hindrance decreasing the drug’s ability to interact with the enzyme.

Combination therapy of afatinib and the EGFR antibody cetuximab given to mice harboring the T790M mutation led to activation of the mTOR pathway and acquired resistance [66]. Two mutations were identified in the mTOR modulating protein NF2 (R198*, a truncating mutation and 811-2A > T, a splicing mutant) (Figure 1C). NF2 encodes the protein Merlin which regulates contact inhibition by several mechanisms including neutralizing growth factor receptors [67,68], and promoting cadherin cell attachments [69]. Contact inhibition is primarily mediated through interaction with CD44 [70,71] or E-cadherin [72] (Figure 1A). In mouse embryonic fibroblasts, loss of NF2 function decreases the strength of cadherin mediated adherens junctions and increases confluency [73]. Merlin is also involved in signaling through the Hippo pathway to restrict growth of tissues [74] and the mTOR pathway [75]. Finally, translocation of merlin to the nucleus indirectly modulates expression of several oncogenes including MYC and RAS [76,77]. Some early reports identified biallelic inactivation of NF2 in mesotheliomas but not lung cancers [78,79] and has been singled as a therapeutic target in these cancers [75]. The NF2 resistant mutations identified in afatinib and cetuximab combination therapy patients [66] show that modulating the mTOR pathway is one mechanism to escape EGFR TK inhibition.

TESEVATINIB (XL647) is another second generation inhibitor [80] that targets both EGFR T790M gatekeeper mutation as well vascular endothelial growth factor (VEGFR) [81], the ephrin receptor B2 [82], and human epidermal growth factor receptor 2 (HER-2) [83]. Patients with Exon19 deletion or the L858R mutation responded well to tesevatinib though its efficacy may be increased because it also targets VEGFR which promotes angiogenesis. However, a subsequent study found that patients who had acquired the T790M mutation after erlotinib and gefitinib treatment had only very modest response to tesevatinib and questioned its efficacy in treatment of patients that acquire this resistant mutation [84]. Patients who only harbored the EGFR activating mutations but were not previously treated with any drug (naïve) fared better [85].

OSIMERTINIB (AZD9291) was developed as a third-generation drug by AstraZeneca to treat patients with the T790M mutation resistant to generation I and II drugs [86,87]. Like afatinib, osimertinib is an irreversible inhibitor that covalently interacts with C797. Unlike afatinib, osimertinib is highly specific for EGFR and does not target the other members of the ErbB1 family. Osimertinib works best against the L858R/T790M EGFR double mutant but not as well as afatinib against L858R single mutant, the Exon 19del mutant or WT EGFR. At least one report found that osimertinib may not be efficient for patients that harbor both the T790M and the exon 19 deletion [88]. As expected, resistant mutations in response to osimertinib treatment arise by inactivating cysteine 797 (C797G or C797S) [89,90] (Figure 1E). A G796D mutation has been reported in a Chinese patient to also be resistant to osimertinib [90]. Molecular modeling shows that this mutation causes steric hindrance that prevents interaction of osmiertinib with C797. Two other mutations G796S/R and L792F/H also interfere with the drug’s ability to interact with the kinase domain [91]. Recently a L718Q mutation was reported to render EGFR resistant to osimertinib [92]. This patient, positive for L858R, acquired the T790M mutation after initial treatment with gefitinib and was subsequently started on osimertinib. Within eight months the patient acquired the L718Q mutation. Remarkably, in vitro analysis suggests that the L718Q T790M double mutant re-sensitizes the receptor to afatinib [93]. Molecular modeling shows that the L718Q mutation appears to block osimertinib from interacting covalently with C797 [94]. Finally, a L747P mutation [95] also renders cells resistant to osimertinib while sensitizing cells to afatinib [96,97]. The L747P single mutation appears to be sufficient to confer resistance to erlotinib or gefitinib [98,99,100] if it occurs spontaneously but does not appear to arise in response to gefitinib/erlotinib treatment. Other mechanisms of resistance including mutations or upregulation of other pathways have also been documented [101,102] but we do not discuss them here.

OLMUTINIB (HM61713) is another third-generation drug [103] that selectively inhibits EGFR T790M, L858R and exon 19 deletions but works poorly against WT EGFR [104,105]. In addition, olmutinib also inhibits the function of several ABC (ATP-binding cassette) transporters [106], which are usually overexpressed in cancer cells showing multidrug resistance [107]. A T790M mutant patient treated with olmutinib acquired the C797S mutation suggesting that the mechanism of action of olmutinib is similar to osimertinib [108]. Several other small molecule EGFR inhibitors have also been synthesized and they are reviewed elsewhere [109,110,111,112].

#### 4.1.3. Hepatocyte Growth Factor Receptor (HFGR, c-Met)

The hepatocyte growth factor receptor HGFR encoded by c-Met is a receptor tyrosine kinase activated by several ligands including the hepatocyte growth factor [113,114]. The signal is transduced through PI3K [115], AKT [116], and mTOR which promotes cell survival [117] (Figure 1A). There is also crosstalk between MET and EFGR or ALK (see Section 4.1.4 and Section 4.3.1) that activates the RAS-RAF-MEK-ERK pathway which promotes cell proliferation [118]. Finally, there are several other MET downstream targets [118] including STAT3 [119].

MET amplification and overexpression occurs often in NSCLC EGFR TK-inhibitors treated patients [120,121,122,123,124,125,126,127]. A mutation that causes alternative splicing of exon 14 known as “exon skipping” leads to an in-frame deletion of a MET negative regulatory domain creating a gain of function allele [128,129] Several other activating mutations have also been identified most importantly in the juxtamembrane domain [130].

Small molecule inhibitors of MET have been generated that target the ATP binding pocket [131,132]. These molecules are known as Class I and Class II. Some but not all Class I inhibitors are highly selective for MET while Class II usually also target other kinases [133,134,135]. Class III inhibitors have also been developed that are non-ATP competitive. Tivatinib is such a molecule that appears to stabilize MET in an inhibitory conformation [136,137,138]. Below we discuss a few examples of resistant mutations that arise against Class I inhibitors.

CRIZOTINIB (PF-02341066) synthesized by Pfizer is a non-selective Class I inhibitor that targets ALK, ROS1 and RON or MST1R (Macrophage stimulating protein receptor) in addition to MET [139] (please see Section 4.1.4 for a discussion on ALK resistant mutations). When bound to MET, crizotinib forms interactions with the activation loop blocking the enzyme in an autoinhibitory conformation [140]. Treatment of patients with crizotinib eventually leads to resistant mutations (Figure 1F). Two mutations in the A loop D1246H/N [141,142,143,144,145] and Y1248H/C/S [141,144,145,146] stabilize the active conformation of the A loop and decreases the efficacy of the drug [147]. At least one study identified the G1181R in addition to the Y1248 and D1246 mutations when an exon 14 skipping patient was treated with crizotinib [145]. G1181R also affects the ATP pocket and is presumed to lead to resistance by the same mechanism.

CAPMATINIB (INC280, INCB28060) was synthesized by Incyte Corporation as a reversible ATP competitive inhibitor highly selective for c-MET [148]. It works best against the exon 14 deletion mutant or the R988C mutant but is also highly potent against the non-mutated allele in cells with MET amplification or over-expression. It interacts with MET through the Y1230 residue. A salt bridge between D1246 and Y1248 further stabilizes the inhibitor [149]. Resistance mutations to capmatinib arise that disrupt the interaction of the drug with the kinase. Patients with MET amplification treated with gefitinib and capmatinib developed the D1246N and the Y1248H resistant mutations [144]. Molecular modeling showed that the mutant c-MET was sufficiently distorted to prohibit the binding of the drug.

SAVOLITINIB (volitinib, AZD6094) was synthesized by AstraZeneca as a Type I specific c-MET inhibitor [150] and shown initially to be active against gastric and renal cancers xenograft models [151,152]. The drug was subsequently shown to be active against NSCLC models both in vivo and in vitro where it also targets PI3K and MAPK [153]. Upon treatment with savolitinib, the D1246V/H/N mutation arises and renders cells resistant to the drug [153] (Figure 1E). Even though savolitinib does not form a molecular interaction with D1246, molecular modeling shows that substitution of the charged *Asp* with uncharged *Val*, *His,* or *Asn* induces a conformation shift in the catalytic A-loop that significantly decreases the affinity of the drug for the enzyme [154,155]. Although we do not discuss Type II or Type III inhibitors in this review, it is worth noting that the D1246V/H/N mutation renders the kinase sensitive to the Type II inhibitors [154]. Savolitinib resistance also arises by over-expression of c-MYC or constitutive expression of mTOR [153].

#### 4.1.4. Anaplastic Lymphoma Kinase (ALK)

ALK is a receptor tyrosine kinase that promotes cell proliferation, survival and migration involved in gut and neuronal development [156,157] (Figure 1A). Although long considered an orphan receptor in vertebrates, ligands have been recently identified in the nervous system [158]. The signal for cell proliferation is transduced through RAS-RAF-MEK-ERK but there is also crosstalk with the PI3K-AKT-mTOR and RAC1/CDC42-PAK pathways [156]. Signal can also be transduced through the JAK3-STAT3 and PLCγ-PIP2-IP3 pathways [159] (not shown in Figure 1). To our knowledge a ligand in lung tissue has not been identified.

ALK disfunction was identified in anaplastic large cell lymphoma as a fusion between the catalytic domain and nucleophosmin (NPM) amino terminus [160]. NPM is a nuclear protein with pleiotropic functions including genome stability and chromatin remodeling [161]. The ALK-NPM fusion appears to allow expression and activity of ALK in lymphatic tissues which contributes to the development of lymphomas. Another fusion between ALK and tropomyosin (TPM) with similar effect as the ALK-NPM fusion has also been described in lymphomas [162]. Fusions between ALK and other proteins also appear to activate ALK in NSCLC [163,164,165,166,167,168]. Most unique to lung cancer is a fusion between ALK and the microtubule associated protein like 4 EML4 [169,170]. Several of these EML4-ALK fusions have been identified in which various EML4 exons are always fused with ALK exons 20–29. All of these fusions lead to ALK gain of function. ALK inhibitors show some efficacy but eventually most acquire resistance [171].

Crizotinib, mentioned above, was one of the first drug used to target NSCLCs characterized by ALK fusions [172]. Crizotinib interacts with ALK similarly to MET through the L1196 gatekeeper residue (L1158 in MET), as well as the G1202 and G1269 residues [173,174]. Other interacting residues are also conserved between MET and ALK but no corresponding tyrosine residue (Y1248 MET) interaction is found in ALK. This loss of interaction appears to be the reason for the higher affinity of the drug for MET than ALK. The drug is effective for about 10 months [172] but eventually resistance develops [175] (Figure 1A). As expected, the most prevalent ALK resistant mutations are L1196M [176,177,178,179] and G1269A [178,179] which disrupt the interaction of the drug with the enzyme and decreases the efficacy of the drug for the EML4-ALK target almost ten-fold [174]. At least two reports have identified the G1202R mutation [177,180] as well as another S1206Y mutation [177,181] which decrease the affinity of the drug for the kinase. Other structural mutations are likely to have the same effect (Appendix A). Two other mutations in residues somewhat distal from the contact residues (C1156Y and F1174V) [179,180,182] show a lower degree of resistance. A G1128A mutation [183] in EML4-ALK lung cancers does not appear to affect the drug’s affinity for the enzyme but rather induces a conformational change that increases the kinase activity [184]. A I1171T/N mutation has also been identified that shows resistance to both crizotinib and the second generation alectinib drug in NSCLC [185].

ALECTINIB (CH5424802) is a second generation selective ALK inhibitor designed to target secondary resistant mutations resulting from crizotinib treatment [186]. Alectinib is very efficient in inhibiting the activity of EML4-ALK L1196M gatekeeper mutant in NSCLC xenograft models [186,187]. An additional advantage of alectinib is that it can penetrate more tissues including the central nervous system (CNS) which is impenetrable to crizotinib [188]. This makes alectinib appealing for treatment of NSCLC which have a tendency to metastasize to the CNS as well as other MET positive tumors. Resistant mutations to alectinib in NSCLC are primarily in the I1171S/N residue mentioned above [185,189,190,191] which cause fluctuations in the A-loop and destabilize the drug ALK interaction [192]. A V1180L resistant mutant [193] has the same properties as I1171S/N. The G1202R resistant mutation resulting from crizotinib treatment also arises in alectinib treatment [171,190].

CERITINIB (LDK378) is another second generation selective ALK inhibitor produced by Novartis with high activity against the ALK L1196M, I1171T, S1206Y and G1269A crizotinib resistance mutations [181]. Similarly to crizotinib, it interacts with the ATP binding pocket of ALK. It can also penetrate the blood brain barrier [194]. Some resistant mutations that decrease the efficacy of ceritinib are similar to those that arise for crizotinib and alectinib [181]. A resistant mutation unique to ceritinib is the G1123S. This mutation and the F1174C/V mutation which also arises in response to crizotinib treatment [181,195] induce a conformation change in the structure of the P-loop to decrease the affinity of the drug [196]. Another unique D1203N mutation was also shown to be resistant in both cell lines and in vivo models [197] although it is currently not clear how it leads to resistance. Due to its proximity to the G1202 residue it can be speculated that it destabilizes the interaction of the drug with the kinase.

### 4.2. Hematopoietic and Lymphoid Tissue

#### 4.2.1. Classification

The World Health Organization (WHO) convenes from time to time to classify tumors of the hematopoietic and lymphoid tissues [198,199]. Both morphology and genetics are considered when classifying hematopoietic and lymphoid tumors. Here we briefly summarize the classification of the tumors discussed in this review.

Chronic myeloid leukemias (CMLs) are generally characterized by the presence of BCR-ABL1 fusions [200]. CMLs can progress to more aggressive forms such as acute myeloid leukemia (AML) or acute lymphoblastic leukemia (ALL). AML and CML are characterized by acquisition of additional mutations to the BCR-ABL1 fusion. Chronic lymphocytic leukemia (CLL) usually develops later in life and is characterized by an increase in lymphocytic count [201]. Genetically, this disease is associated with BCR mutations and several chromosomal deletions including 13q14, 11q22-23 (associated with ATM: ataxia telangiectasia mutated, a checkpoint gene), SF3B1 (splicing factor 3b subunit 1), and BIRC3 (baculoviral IAP repeat containing 3) mutations [202,203,204] and 17p13 (associated with TP53 mutations) as well as trisomy 12. Genomic sequencing identified several other altered pathways including NOTCH1, apoptosis, and cell cycle defects [205]. Lymphoplasmacytic lymphoma (LPL) is an elderly rare disorder, a subtype of non-Hodgkin’s lymphoma, genetically characterized by mutations in MYD88 and CXCR4 [206]. Another form of non-Hodgkin’s lymphoma is mantle cell lymphoma (MCL) which overexpresses Cyclin D1 and also shows defects in TP53, NOTCH1/2 PI3K, mTOR and NF-kB in more aggressive forms [207].

#### 4.2.2. Abelson Murine Leukemia Viral Oncogene Homolog 1 (ABL1)

ABL1 is an ubiquitously expressed non-receptor tyrosine kinase but with highly controlled enzymatic activity [208] with several functions including cell differentiation, division and adhesion [209,210,211,212] (Figure 2A).

Interaction between the SH3 domain and the kinase domain forms an inhibitory conformation [213]. SH3 domain mutations serve as activating c-Abl1 mutations in various hematopoietic and lymphoid cancers. However, a hallmark of many leukemias is the presence of the Philadelphia chromosome, a reciprocal translocation between chromosome 22 and 9 [214,215,216,217]. The break in chromosome 22 occurs within the BCR gene at one of three different locations: just after the first exon, around exon 13 and after exon 19. Nearly the entire Abl1 gene is fused to one of these three fragments of BCR creating three different variations (p190^Bcr-Abl^, exon 18 BCR break; p210^Bcr-Abl^, exon 13 BCR break; and p230^Bcl-Abl^, exon 1 break) [218]. WT BCR and ABL1 can interact with each other and it appears that BCR1 negatively regulates ABL1 [219]. Remarkably, the BCR-ABL fusion permanently activates the ABL tyrosine kinase activity [220]. Pathways destabilized by the BCR-ABL fusion include cell cycle regulation through Ras signaling, apoptosis [221,222,223], cell adhesion [224,225], loss of growth factor interaction [226,227], and DNA repair (NER and DSB repair) (Figure 2A) [228,229].

IMATINIB MESYLATE (STI571) is the first-generation drug designed by Novartis Pharma as a specific PDGF-R inhibitor. The subsequent realization that it selectively inhibits both Abl1 and Bcr-Abl kinase activity facilitated FDA approval for CML treatment [230,231]. Imatinib is a strong specific inhibitor of ATP binding to Bcr-Abl [232]. Crystal structure of the Abl kinase domain and imatinib complex reveals that a hydrogen bond between the drug and T315 (the gatekeeper residue) stabilizes the drug in a conformation that blocks ATP interaction with the enzyme active site [233,234,235].

Resistance to imatinib eventually develops [236,237]. The T315I gatekeeper mutation is by far the most commonly occurring mutation that renders cells resistant to Imatinib [238,239] (Figure 2B, Appendix A). Substitution of threonine for isoleucine abrogates the hydrogen bond required for Imatinib interaction and drastically reduces the affinity of the drug for the enzyme [240]. The F311I/L and F317L/V also affect the drug affinity, but they are not as drastic because they do not completely abrogate the interaction (Appendix A).

Remarkably several other mutations have also been reported to render resistance to imatinib. In order of incidence reported on COSMIC they are E255K/V > Y253F/H > G250E > M351T > F359A/C/I/V/X > M244V > F317L/V (mentioned above) > E355A/G/X > Q252E/H/K/R > E459G/K/Q, H396P/R. Others arise at much lower frequency (Appendix A). The Y253F/H, E255K/V, G250E, M244V, and Q252E/E/H/K/R detected in the BCR-ABL patients are all found in the P-loop of the kinase [241]. The P-loop is the ATP binding site and these mutations generally reduces the drug’s affinity for the enzyme but does not affect the kinase activity [242,243,244]. It has been demonstrated that the sensitivity of the kinase to imatinib is decreased in patients harboring these mutations by 70%–100% [245]. At least one of the P-loop mutations (Y253F) appears to also act as an activating mutation because it increases the enzyme activity compared to WT [246]. The M351T, F359A/C/I/V/X, and E355A/G/X are in the SH2 contact region (Appendix A). The SH2 and SH3 interactions with the kinase domain promote an inhibitory conformation that inactivate the enzyme [247,248,249]. These and other mutations in the regions where the SH2 or SH3 domains contact the kinase domain (Appendix A) affect the autoinhibitory conformation form of the enzyme and decrease the drug’s potency. The H396P/R mutation is in the activation (A)-loop of the enzyme. It appears to stabilize the loop from a closed to an open conformation causing the enzyme to remain activated for longer periods of time [250].

DASATINIB, nilotinib and bosutinib are second generation BCR-ABL inhibitors [251,252,253,254]. Dasatinib (BMS-354825) was designed by Bristol-Myers Squibb and Otskuka Pharmaceutical Co., Ltd. to inhibit imatinib resistant BCR-ABL mutations in CML but it also works against Src family kinases [255,256,257,258]. The crystal structure of the kinase domain + dasatinib complex shows that the drug makes hydrogen bonds with T315 and M318. It efficiently inhibits most imatinib resistant mutants except the T315I gatekeeper mutation. In fact, if patients without the T315I mutation are treated with dasatinib they also acquire this resistant mutation as well as the neighboring F317L/R (Appendix A). A few other mutations most importantly in the P-loop (E255K Y253H and G250E) also eventually confer resistance to the drug. These mutations are similar to the imatinib resistant mutations and confer resistance by a similar mechanism (Appendix A) [259].

NILOTINIB (AMN107) was designed by Novartis Pharmaceuticals to inhibit Abl kinase activity in imatinib-resistant cells [260,261,262,263]. The crystal structure of the Abl-Nilotinib complex shows that although it occupies the same hydrophobic pocket as imatinib, nilotinib fits better in this pocket and therefore has higher affinity for the enzyme making it a more potent inhibitor [263]. However, the drug is not very efficient against P-loop mutations (Y253F/H, E255K/V) that affect the inhibitory conformation of the enzyme or the T315I gatekeeper mutation with which it makes a hydrogen bond similarly to imatinib and dasatinib. In fact, nilotinib resistant secondary mutations arise in the same amino acids (P-loop, T315, SH2 contact site) as for imatinib and dasatinib [264] (Appendix A)

An advantage of nilotinib is that it also inhibits KIT, both WT and the D816G/V mutant resistant to crizotinib (see Section 4.3.1), making it useful for treatment of other *c-Kit* driven tumors [263,265,266]. Thus, nilotinib is used for treatment of advanced gastro-intestinal tumors (GISTs) [267]. A resistant KIT N655T mutation was identified in a treated GIST patient [268]. This mutation also appears to arise spontaneously [269].

BOSUTINIB (SKI-606) was subsequently produced by Wyeth (now Pfizer Inc., NY, USA) as a Bcr-Abl ATP competitive inhibitor with some activity against Src [270,271]. Bosutinib can inhibit BCR-ABL at nanomolar range and is less toxic than imatinib, nilotinib and dasatinib [272,273,274,275,276] making it a more desirable drug. The drug’s therapeutic potential is also enhanced by its dual inhibition of Src and Abl [277]. The crystal structure of bosutinib complexed with the enzyme shows that its interaction is similar to that of dasatinib with the exception that it only forms a hydrogen bond with M318 and only Van der Waals interactions with T315 [278]. In addition, bosutinib also does not interact with the P-loop making it good for treating P-loop resistant mutations compared to the other drugs. Further, unlike nilotinib and dasatinib, bosutinib can inhibit all but the T315I and V299L imatinib resistant mutants [271,277,279]. Even though the drug forms no interaction with the T315 and V299 residues, the T315I and V299L substitutions causes sufficient steric hindrance that drastically decreases the affinity of the drug for the enzyme [280]. An L248R mutation in the P-loop which was previously shown in vitro to cause imatinib resistance [281] was clinically isolated from a F359I imatinib resistant patient subsequently treated with bosutinib [282]. In vitro analysis showed that although the L248R single mutation confers only moderate resistance than the L248R F359I double mutant it is highly resistant to bosutinib as well as the other drugs. Because bosutinib does not interact with the P-loop, the L248R resistant mutation is surprising and to our knowledge no in silico modeling with this mutation has been performed. Except for the Y253F mutation, most P-loop mutations decrease the affinity of the drug for the enzyme. Mutations in the F359 residue alter the inhibitory conformation of the enzyme. Therefore, it is possible that the L248R F359I double mutant may cause resistance through a combination of decreased drug affinity and increase enzymatic activity.

#### 4.2.3. Bruton’s Tyrosine Kinase (BTK)

The cytoplasmic Bruton’s tyrosine kinase (BTK) is a member of a complex B-cell signal transduction relay required for survival, proliferation, migration and apoptosis of B-cells [283,284]. Interaction of an antigen with the B-cell receptor (BCR) activates a cascade of signal transduction pathways (only a simplified form shown in Figure 2A) that eventually leads to activation of BTK [285,286,287,288]. BTK phosphorylation at Y551 in response to BCR signaling significantly increases the kinase activity of the enzyme [289,290,291,292]. BTK may also autoactivate by self-phosphorylation of Tyr223 in the SH3 domain which appears to increase its affinity for downstream targets [289,293,294,295,296,297]. Activated BTK promotes B-cell maturation.

Mutations in BTK that affect its enzymatic activity have initially been identified in inherited X-linked agammaglobulinemia (XLA) [298]. XLA patients have a deficiency in B-cell maturation resulting in very low plasma B-cell counts [299]. Constitutive BTK activation is essential for proliferation of abnormal B-cells in several types of blood malignancies including chronic lymphocytic leukemia (CLL) [300,301], mantle cell lymphoma [302,303], and lymphoplasmacytic lymphoma [304]. Remarkably, activating oncogenic BTK mutations have not been clinically identified. The kinase appears to be activated by other mutations in the components of the BCR signaling pathway. Mutations in ITAM, CD79A, CD79B [305] as well as other members of the BCR signal transduction pathway [283] have been reported in various cancers. Nevertheless, inhibition of BTK or other components of the BCR signaling pathway drastically reduces B-cell proliferation in vitro [305] making BTK an attractive druggable target.

IBRUTINIB (PCI-32765) was the first generation targeted BTK inhibitor with subsequent development of others (Acalabrutinib, Tirabrutinib, Zanabrutinib, Spebrutinib) [306]. A team at Celera Genomics synthesized the irreversible inhibitor ibrutinib in 2006 [307]. The inhibitor forms a covalent bond with C481 and inhibits the kinase activity of BTK and autophosphorylation at Y223 [308,309,310].

Resistance to ibrutinib develops almost always by amino acid substitution of C481S which abrogates the covalent bond of the inhibitor with the kinase (Figure 2D). This mutation restores the BTK Y233 autophosphorylation and the kinase activity for downstream targets [311,312,313]. Remarkably, ibrutinib may still inhibit the C481S mutant form but only reversibly and about 100 times less efficiently. The Abl inhibitor, dasatinib which is a less effective WT BTK inhibitor than ibrutinib inhibits both the mutant and non-mutant form with the same effectiveness [312]. C481F/R/Y substitutions have also been reported (Figure 2D, Appendix A). A clinically identified T316A resistant mutation in the SH2 domain has the same effect as the C481S mutation [314]. Molecular modeling shows that this mutation does not affect ibrutinib interaction with the enzyme. Nevertheless, both Y223 autophosphorylation and activation of downstream targets were comparably decreased in C481S and T316A.

#### 4.2.4. FMS-like Tyrosine Kinase 3 (FLT3)

FLT3 is a receptor tyrosine kinase that appears to be expressed mainly in hematopoietic cells [315,316]. It is a Type III receptor tyrosine kinase (RTK) that has a distinct activation mechanism [317]. Interaction of the juxtamembrane domain with the kinase domain leads to an inactive conformation state [318]. Upon ligand binding, a conformation shift occurs that cross-phosphorylates and activates the kinase. This activation mechanism is unlike other RTKs where ligand binding dimerizes the receptor and causes cross-phosphorylation of activation loop [319]. Consequently, FLT3 mutations or alterations of the juxtamembrane domain constitutively activate the receptor [320]. One such activating change identified in acute myeloid leukemia is a tandem duplication of the juxtamembrane domain that affects the inhibitory conformation of the enzyme and leads to constitutive activation [321,322,323,324,325]. Further, in vitro analysis has shown that shortening of the juxtamembrane domain also leads to constitutive activation of the enzyme [320]. Less frequently, two other activation mutations in D835 and I836 have been clinically identified in AML patients [326]. These mutations are in the activation loop of the enzyme and are able to induce an active conformation change [327,328]. Physiologically, the function of FLT3 is to promote survival and proliferation which is accomplished by activating several parallel pathways (Figure 2A) [329]. Several anti-FLT3 drugs have been developed for the treatment of AML patients [329,330]. Here we only discuss resistant mutations to sorafenib, quizartinib, and sunitinib in AML patients.

SORAFENIB (BAY 43-9006) was initially identified through a collaboration of Bayer and Onyx Pharmaceuticals in a screen for Raf1 inhibitors. Subsequent structural analysis allowed minor modifications of the chemical compound to increase its Raf1 inhibitory potency [331,332,333,334]. In addition to Raf1, sorafenib can also inhibit b-Raf, BEGFR1/2/3, PDGFR, FGFR1, c-kit, FLT3, and RET making it a broad-spectrum drug [335,336]. Following clinical trials, the drug was approved to treat various cancers including AML [337]. The most often detected resistant mutation to sorafenib is in the D835 residue mentioned above which is often substituted for other residues including H, Y, and X [338,339] (Figure 2C, Appendix A). Molecular modeling shows that D835 which is in the activation loop is crucial for facilitating docking of the drug with the enzyme [327,340]. Any mutation in this amino acid severely affects the ability of the drug to inhibit FLT3. An F691L gatekeeper mutation has also been identified in sorafenib treated patients (Figure 2C, Appendix A) [339]. Molecular modeling shows that sorafenib makes contact with this residue and the F691L change prohibits this contact [339].

QUIZARTINIB (AC220) was identified in a screen for specific FLT3 inhibitors [341,342]. The drug has high selectivity for FLT3 with minor effects on other kinases and has been used for treatment of AML [343,344]. Crystal structure of the FLT3 catalytic domain bound to the drug shows that similarly to sorafenib it makes interactions with F691 but not D835 [345]. Predictably, the F691L resistant mutation affects the drug’s ability to interact with the enzyme while the D835 mutations cause a hyperactive kinase form [327,339,346,347].

SUNITINIB (SU011248) is another drug that targets several receptor tyrosine kinases including KIT and FLT3 [348]. It was initially approved for treatment of gastrointestinal tumors [349]. Unlike quizartinib and sorafenib, sunitinib does not make interactions with F691 [339] and consequently F691L resistant mutations have not been identified. However, a D835Y mutation affects the ability of the drug to interact with the enzyme similarly to sorafenib and quizartinib.

For a more in depth reading of skin cancer resistance to targeted therapies there are several recent reviews [350,351].

### 4.3. Gastrointestinal Stromal Tumors (GISTs)

Gastrointestinal stromal tumors (GISTs) occur in the cells of Cajal which coordinate smooth muscle contractions in the gastrointestinal tract [352,353,354,355]. GISTs show c-KIT and PDGFRA gain of function, two receptor tyrosine kinases that promote the cell cycle and cell proliferation (Figure 3A) [356,357].

#### 4.3.1. Tyrosine-Protein Kinase KIT

KIT is a receptor tyrosine kinase which appears to function primarily in stem cells [358]. It has also been characterized as a dependence receptor which can function in the absence of a ligand to trigger apoptosis [359]. KIT signals through the PI3K-AKT-mTOR pathway to promote anti-apoptosis and through the RAS-RAF-MEK-MAPK pathway to activate proliferation (Figure 3A) [360,361]. Activating *c-KIT* oncogenic mutations have been first identified in gastrointestinal and leukemia cancers [362,363,364] as well as a variety of other hematopoietic cancers [365,366]. Exon 11 encodes the autoinhibitory juxtamembrane domain where many KIT mutations are found [360,367]. Another common D816V mutation occurs in the activation loop [368,369]. Caspase dependent cleavage of c-KIT at D816 is required for its pro-apoptotic activity and the D816V prevents cleavage [359].

#### 4.3.2. Platelet Derived Growth Factor Receptor Alpha (PDGFRA)

PDGFRA is a receptor tyrosine kinase that signals through the same pathways as KIT (Figure 3A) [370]. About 15% of GISTs patients have activating mutations in PGFRA and not KIT. PDGFRA mutations occur in the juxtamembrane domain but also in the activation loop and ATP binding domain [357]. The D842V mutation is the most common which occurs in one of the tyrosine kinase domains [371].

Imatinib, nilotininb and sunitinib are all used for treatment of GISTs. Imatinib is able to inhibit KIT [372] and PDGFR [373] receptors. KIT D816 mutations (D816A/E/G/H) also appear as secondary mutations in response to the drug (Figure 3B, Appendix A). The two other most reported KIT resistant mutation to imatinib are V654A and Y823D. Molecular modeling shows that these mutations decrease the affinity of the drug for KIT [374]. In PDGFRA, the D842V substitution occurs most often (Figure 3C, Appendix A) [375]. Structural analysis shows that the D842V is an activating mutation that increases PDGFRA affinity for ATP [376]. A melanoma patient treated with imatinib developed both a L576P KIT mutation and a CTNNB1 S33C mutation (Figure 3D) [377]. Modeling simulations show that the L576P KIT mutation which also appears in GIST [378] is an activating mutation [379]. CTNNB1 encodes beta-catenin which is part of the Wnt signaling pathway that has diverse functions including cell proliferation and migration [380,381]. The S33C mutation knocks out a phosphorylation site and appears to stabilize the protein [382,383].

A D816G KIT mutation has also been recently identified in a NSCLC patient treated with crizotinib [384]. This patient also harbored a ROS1 chromosomal re-arrangement. The ROS1 protooncogene is a receptor tyrosine kinase similar to ALK [385,386] and re-arrangements of ROS1 have been described as “driver mutations” in lung cancers [387,388,389,390,391]. Tyrosine Kinase inhibitors including crizotinib and ceritinib have been used for treatment of lung cancers harboring ROS1 re-arrangements [392,393,394]. Treatment of this ROS1 re-arranged patient with crizotinib for 15 months led to acquired resistance to the drug. Subsequent sequencing revealed that the KIT D816G mutation causes the resistant phenotype. In vitro assays showed that this mutation induces KIT autophosphorylation and activation. These data suggest that activation of the KIT pathway bypasses the inhibition of the ROS pathway.

### 4.4. Melanoma

#### Classification

Melanomas are malignant melanocytic neoplasms. Although malignancies can occur in any tissue where melanocytes are found, skin melanomas are the most prevalent in sun exposed white people [395,396]. Skin melanomas are generally classified as chronically sun damaged (CSD) and non-chronically sun damaged (non-CSD) [397]. CSD melanomas are characterized by high mutation burden that leads to disruption of several pathways [396]. The serine/threonine kinase BRAF^V600E^ mutation that disrupts the KIT-NRAS-BRAF-MEK-ERK-CDK4/6-RB pathway (Figure 4) is almost always present in both CSD and non-CSD melanomas [398]. The mutation also appears in NSCLC [399]. The CSD melanomas often have additional KIT, NRAS and NF1 mutations [400]. The BRAF^V600E^ is a driver mutation in melanomas because it constitutively activates BRAF and the ERK pathway [401,402].

The KIT receptor tyrosine kinase described above is also mutated in melanoma [400,403]. Shain and colleagues [404] sequenced 293 relevant genes in melanoma tissues and delineated the progression of mutations in these malignancies. They showed that BRAF^V600E^ is almost always the first mutation to occur followed by NRAS and TERT (telomerase). Homozygous CDKN2A, PTEN and TP53 deletions are associated with invasive and metastatic melanoma. Mutations in the neurofibromatosis factor NF1 are also associated with invasive and metastatic melanomas [404,405]. One study showed that NF1 mutations suppress BRAF targeted therapy making BRAF inhibitors less efficient against BRAF and NF1 double mutants [406]. The therapeutic agents used for melanoma treatment have been reviewed recently [407]. Below we discuss resistant mutations that arise in response to some of these agents.

VEMURAFENIB (PLX4032) is a BRAF^V600E^ inhibitor discovered by researchers at Plexxikon [408,409]. Drug synthesis was informed through a crystallography and molecular modeling approach. Because previous BRAF drugs were not very specific [410,411], the investigators wanted a compound that targeted BRAF^V600E^ but had little activity against WT BRAF or other kinases. This method allowed synthesis of highly selective compounds including PLX4720, another BRAF^V600E^ not discussed here [408]. Vemurafenib forms a hydrogen bond with Asp594. A salt bridge between E600 and L507 prevents the activation loop from interacting with the ATP binding site, thus inhibiting enzyme activation. Preclinical trials showed that vemurafenib is a very effective inhibitor [409,412]. Subsequent clinical trials showed that vemurafenib induces efficient tumor regression in BRAF^V600E^ melanoma patients as well as other cancers [413,414,415,416,417]. Vemurafenib and the related drug dabrafenib activate WT BRAF and are only indicated for BRAF^V600E^ patients [418].

Vemurafenib resistant mutations arise in several genes in the MAPK pathway including MEK1 (MAP2K1), MEK2 (MAP2K2), NRAS, PIK3CA, and PTEN as well as various alterations of the BRAF gene itself (Figure 4, Appendix A) [419]. The MEK1 C121S mutation leads to a hyperactive form of the enzyme which remains active in the absence of an upstream signal [420]. A screen for BRAF inhibitor resistant mutations identified the MEK1 G128V, P124L/S, and V60E mutations as well as other MEK2 and NRAS [421]. Various BRAF splice variants also confer resistance to vemurafenib [422] (Appendix A). These variants usually lack the RAS binding site which resides between exons 4–8 and allows BRAF activation in the absence of RAS interaction [423,424,425,426]. A BRAF L505H mutation [427,428] appears to affect the ability of the drug to prohibit activation loop interaction with the binding site (Figure 4B). Remarkably, molecular modeling showed that the double V600E L505H mutant may still be sensitive to sorafenib [427]. Finally activating mutations in PIK3CA [429,430] or complete inactivation of PTEN [423] also arise in response to vemurafenib.

DABRAFENIB (GSK2118436) is an inhibitor that interacts with the inactive conformation of BRAF^V600E^ [431,432,433,434,435,436]. It was synthesized by GlaxoSmithKline [437]. X-ray structure of dabrafenib complexed with BRAF^V600E^ shows that the drug is stabilized in the active site of the enzyme by hydrogen bonds with C532, K483, and F595 [433]. Several other hydrophobic interactions are required for drug interaction with the active site of BRAF^V600E^. In addition to BRAF^V600E^, dabrafenib also inhibits MEK (MAP2K1) and MEK2 (MAP2K2) [438] and NRAS [431].

BRAF^V600E^ resistance to dabrafenib has been reported. In one case, a splicing variant missing exons 2–10 was isolated from dabrafenib treated patients (Appendix A) [422,423,439]. This splicing variant was initially described as a resistant mutation to vemurafenib treated patients [426]. Deletion of this region causes BRAF dimerization and activation in the absence of RAS interaction. Other BRAF structural alterations that confer resistance to vemurafenib and dabrafenib have also been identified (Appendix A). BRAF amplification, MEK1/2 and NRAS mutations are other forms of BRAF^V600E^ acquired resistance to dabrafenib [440]. Identified MEK1 resistant mutations are V60E, P124S, C121S, G128V, K57E, and G128D (Figure 4C) [421,422,441]. Most of these mutations (with the exception of V60E and K57E) map in the protein kinase domain. Western blotting shows that the V60E and G128V mutations cause hyperphosphorylation of MEK1/2 and ERK1/2. Therefore, these mutations appear to constitutively activate the MEK/ERK pathway. Dabrafenib MEK2 resistant mutations arise in amino acids V35M, L46F, C125S, and N126D (Figure 4D) [421]. Two of these mutations (V35M and L46F) occur in the N-terminus while two others (C125S and N126D) in the kinase domain. The mutations flank the ATP binding pocket and all cause hyperphosphorylation of MEK1/2 and ERK1/2. Combination therapy of dabrafenib and other inhibitors of the MAPK pathway (e.g., trametinib a MEK inhibitor) [442] leads to mutation in other genes in the pathway such as MEK2 or NRAS [443]. For a more comprehensive review of MAPK pathway resistant mutations please see [444].

SELUMETINIB (ARRY-142889, AZD6244) is a second generation MEK1/2 inhibitor synthesized by AstraZeneca [445,446]. The drug effectively inhibits purified MEK1’s ability to phosphorylate ERK1 as well as tumor growth in mouse xenografts [445]. It is also better at inhibiting tumor cell line growth that harbor BRAF or RAS mutations [446]. Following clinical trials, the drug has been used for the treatment of melanoma [447,448]. The drug is also used for the treatment of NSCLC [449,450] where it appears to work best in combination with PIK3CA inhibitors [451,452] as well as other cancers not discussed here. Cells resistant to selumetinib also show resistance to vemurafenib suggesting that resistance is acquired by the same mechanism [453]. The MAP2K1 P124L mutation was identified in one patient with advanced melanoma treated with selumetinib (Figure 4C) [454].

PEMBROLIZUMAB (MK-3475), designed by Merck &Co, is a monoclonal antibody that targets the program cell death-1 (PD-1) receptor [455]. It was approved in the US for the treatment of NSCLC, breast cancer, lymphoma, melanoma, as well as other forms of cancers [456]. PD-1 is expressed in the T-cells and interacts with the PD-L1 and PD-L2 ligands on the antigen-presenting cells [457]. This interaction suppresses the immune system. Thus PD-1 and other receptors like it act as an immune checkpoint to prevents an autoimmune response [458,459]. Binding of pembrolizumab to PD-1 blocks the receptor–ligand interaction and allows the immune system to attack cancer cells [460]. Although we do not discuss them here, nivolumab is another anti PD-1 antibody, while atezolizumab, avelumab, and durvalumab are anti PD-L1 antibodies that have been FDA approved [461,462]. Ipilimumab [463] is an antibody against CTLA-4 another inhibitory receptor [464,465]. Resistance to pembrolizumab arises by inactivating the function of the JAK1 and JAK2 genes [466]. JAK1/2 are involved in interferon signaling [467,468]. An in-depth review of PD-1/PD-L1 mechanisms of resistance mutations has been recently published [469].

VISMODEGIB (GDC-0449) was discovered by scientists at Genetech in a screen for inhibitors of the hedgehog signaling pathway [470]. It targets SMO (smoothened), a transmembrane protein that transduces signals to the cytoplasm upon binding to cholesterol [471,472,473]. The hedgehog pathway facilitates embryonic development [474] but is also activated in cancer where it contributes to cellular proliferation and differentiation (Figure 4A) [475,476,477]. Hedgehog pathway signaling activates the glioblastoma Gli1/2/3 genes (named so because they were initially identified in glioblastomas [478]) through a complex epistatic pathway of negative regulators [479]. Targeting of the hedgehog pathway in cancer cells has been an efficient way to inhibit tumor growth [480,481,482,483].

Vismodegib has been useful for treating metastatic tumors [484,485,486] including advanced basal cell carcinoma [486,487,488]. Resistance can be inherited or acquired de novo. Whole exome sequencing of vismodegib resistant basal cell carcinomas from patients with Gorlin syndrome, an autosomal dominant disease that causes early onset basal cell carcinoma, identified several mutations in SMO [489,490]. Several other mutations were identified in basal cell carcinoma patients not previously diagnosed with Gorlin syndrome (Figure 4E) [490]. Molecular modeling and docking on a crystal structure of the SMO transmembrane domain [491] showed that resistant mutations fall into two categories, those that affect drug interaction with SMO and those that lead to constitutive activation of SMO. Most mutations occur in the transmembrane domain of the protein.

### 4.5. ESR1 in Breast Cancer

#### Classification

Breast cancer is classified genetically by the status of three receptors: Estrogen receptor (ER), human epidermal growth factor receptor (HER2), and progesterone receptor (PR) [492,493]. Cancers that do not express any of the receptors are known as “triple negative”. Around 70% of all breast cancers are ER-positive and are generally treated with anti-hormone (endocrine) therapy [493]. Two isoforms of the estrogen receptor exist: ERalpha (ESR1) and ERBeta (ESR2) [494]. Upon binding to estrogen (the ligand), the receptor dimerizes and translocate to the nucleus [495] (Figure 5A).

The receptor functions as a homodimer and while ERα/ERβ heterodimers have been reported in vitro, they have not been identified in vivo. In the nucleus, the dimers bind to the estrogen response elements (EREs) [497,498]. Binding of the ligand does not appear to be required for ESR1-ERE interaction in vitro but it strongly increases its affinity in vivo [499,500]. ESR1 phosphorylation further increases its affinity for ERE [501]. In the nucleus, it activates estrogen responsive genes including TP53 [502], BRCA1 [503], CDK4/6 [504], as well as many others [505,506,507,508,509]. These genes promote cellular proliferation and survival [506].

ENDOCRINE THERAPY refers to modulation of hormone activity or complete hormone removal to treat breast cancer [510]. This can be accomplished either by preventing the ovaries from making hormones or by interfering with the function of estrogen and progesterone. Pre-menopausal patients who still produce estrogen in the ovaries have traditionally been treated by surgically removing the ovaries though endocrine therapy has decreased this practice in recent years [511,512,513,514]. Resistant mutations arising from endocrine therapy treatment have been thoroughly reviewed recently [515,516]. Generally, activating mutations in ESR1 renders the receptor resistant to endocrine therapy [517].

TAMOXIFEN. The selective estrogen receptor modulator (SERM) Tamoxifen (ICI46,474) was discovered by ICI Pharmaceuticals Division and has quickly been adopted as a treatment option [518,519] for both pre- and post-menopausal women. Tamoxifen is metabolized to 4-hydroxytamoxifen (OHT) which is the active antagonist that interreacts with the ligand binding domain (LBD) of ERα [520]. The crystal structure of the LBD-OHD complex shows that the drug blocks interaction of ESR1 with its co-activator [521,522]. Helix 12 of the ligand binding domain is required for interaction of ESR1 with co-activators and tamoxifen induces a conformation change that prevents this interaction. Molecular modeling shows that a D538G amino acid substitution frequently identified in tamoxifen treated patients is an ESR1 activating mutation that makes the ligand dispensable (Figure 5B, Appendix A) [523]. A Y537S/N/C mutation increases the flexibility of helix 12 by restricting the number of stabilizing hydrogen bonds formed and also allows interaction of ESR1 with its co-activators in the absence of the ligand [523,524]. Combination therapy is suggested for patients with ESR1 activating mutations which include CDK4/6 or mTOR inhibitors [525,526,527,528].

Aromatase (CYP19A1) is the enzyme responsible for synthesizing estrogen from androgen precursors [529]. Aromatase inhibitors and gonadotropin releasing hormone agonists suppress estrogen production and are often prescribed as adjuvant therapy to post-menopausal patients [530,531,532]. The ESR1 D538G, Y537S/C/N, and to a lesser degree L536H/R activating mutations also appear in aromatase inhibitor treated patients (Figure 5B, Appendix A) [529,533,534,535]. Two other mutations E380Q and S436P arise at significant rate in aromatase inhibitor treated patients that cause constitutive ER activity in the absence of ligand [536,537,538,539]. Finally, several other mutations have also been identified more recently [536,537,540,541]. An E380Q mutation found in helix 12 next to the estrogen binding site increases the flexibility of this helix and decreases the affinity of the protein for the drug [542]. Not discussed here are resistant mutations in aromatase (CYP19A1).

### 4.6. Androgen Receptor in Prostate Cancer

The androgen receptor (AR) is a transcription factor that controls male developmental genes [543,544,545]. Upon interaction with its ligand, the testosterone derivative 5-alpha-dihydrotestosterone (DHT), the androgen receptor binds androgen-responsive elements (AREs) [546,547] (Figure 6A).

The prostate specific antigen (PSA) is an AR target. For full activation of PSA transcription, the AR must interact with both the promoter region and an enhancer regulatory cassette [548]. The proliferation of prostate cancer cells requires constitutive AR activation and one of the treatments is androgen ablation which constitutes blocking production of AR binding ligands (male hormones) [549,550]. For example, luteinizing hormone releasing hormone (LHRH) agonists decrease the levels of androgen receptors produced by the testicles and are used as a form of treatment [551]. Surgical castration has also been shown to be effective because AR receptors originate in the testis [552]. However, ablation provides only temporal relief because prostate cancer cells often respond by over-expressing AR [552,553,554,555,556]. Mutations have also been identified which increase the transcription factor activation potential making it more likely to promote the same transcriptional levels even when ligands are depleted. Some of these mutations appear only after androgen ablation while others after subsequent treatment with therapeutic agents [553,557,558,559,560,561,562,563,564].

Prostate cancer resistant to castration eventually develops in almost 100% of the cases. Therefore, therapeutic agents must be administered for efficient treatment [565]. The two major types of treatments for castration resistant prostate cancer are inhibitors of synthesis of androgen ligand precursor (ketoconazole and abiraterone) and AR antagonists (enzalutamide and flutamide). Even in castrated patients, androgen ligands may still be synthesized by other pathways or de novo. Both ketoconazole and abiraterone are inhibitors of CYP17A [566,567] (Figure 6A), a hydroxylase/lyase involved in steroid synthesis that appears to be responsible for continual androgen synthesis in castrated patients [568]. Ketoconazole is significantly more toxic that abiraterone [569,570]. Abiraterone was developed later by modification of esters of pyridyl acetic acid [571,572], a potent inhibitor of CYP17A [573]. It is a more selective inhibitor than ketoconazole [571,574,575]. The antagonists enzalutamide and flutamide block androgen binding to AR significantly reducing ARs ability to activate transcription [576,577,578,579,580].

Resistance mutations to the AR drugs are generally restricted to two amino acids, 877 and 878 (Figure 6B, Appendix A). The most frequent mutation is T878A which can confer resistance to all drugs mentioned above [581,582,583,584]. T878S was identified in at least one abiraterone resistant cancer [581,584] and F877L in one enzalutamide treated cancer [583,585]. Modeling data of the AR ligand binding domain shows that these mutations are selected for because they convert the antagonists enzalutamide and to a lesser degree flutamide into agonists [586]. Treatment with ketoconazole and abiraterone causes upregulation of AR synthesis but with remarkable selection for the mutant form (T878A) [582]. Importantly, none of the patients that upregulate AR T878A were previously treated with flutamide or enzalutamide. Thus, the mutant can occur directly in response to ketoconazole and abiraterone.

### 4.7. Other Resistant Mutations

INFIGRATINIB (BGJ 398) is a drug that can inhibit the function of fibroblast growth factor receptors 1–4 (FGFR 1–4) [587,588]. The FGFR receptors are good targets because they are involved in a cascade of signaling pathways that control proliferation, cell survival, and angiogenesis [589,590,591]. Infigratinib is able to inhibit FGFR function in sorafenib resistant cells [592] and is efficient in treatment of biliary tract cancer [593,594]. A secondary resistant mutation (V564) to infigratinib has been identified in the FGFR2 gene [595]. Molecular modeling shows that the V564F is a gatekeeper mutation that blocks the drug interaction with the kinase while retaining the activity of the enzyme (Appendix A).

EVEROLIMUS (RAD001) is a rapamycin analog that targets mTOR [596]. Everolimus is used as a chemotherapeutic agent for a variety of cancers including thyroid cancer. A patient with anaplastic thyroid cancer treated with everolimus acquired the mTOR F2108L secondary resistant mutation (Appendix A) [597]. This mutation is similar to a rapamycin resistant mutation identified in fission yeast *tor2* (F2049L) [598]. Structural modeling showed that the human F2108L occurs in the rapamycin binding domain. Subsequent in vitro studies showed that this mutation also affects the effect of rapamycin on mTOR activity. In at least one report, a breast cancer cell line treated with the mTOR inhibitor rapamycin led to the F2108L resistant mutation [527]. Molecular modeling showed that this mutation inhibits drug interaction with the protein [597]. Thus, the mechanism of everolimus resistance is likely to be similar to that of rapamycin.

PF-04217903 is a c-Met ATP competitive inhibitor that can slow or stop tumor progression both in vitro and in vivo [599]. A patient with a kidney tumor was treated with PF-04217903 in a Phase I clinical trial acquired a M1268T secondary resistant mutation (Appendix A) [600]. The significance of this mutation remains to be investigated.

PD0325901, developed by Pfizer, is a non-competitive ATP inhibitor of MAP2K1/MAP2K2 [601,602]. PD0325901 and similar MAP2K specific inhibitors can be used to target the ERK pathway in a variety of cancers [603,604]. F129L and L115P MAP2K1 resistant mutations arise after drug treatment (Appendix A) [605]. These mutations appear to affect drug affinity for the kinase. The F129L also seems to function as an activating mutation [606]. A V215E MAP2K2 mutation has the same effect as the F129L MAP2K1 mutation.

## 5. Conclusions and Future Directions

Resistance to small molecule inhibitors highlights the plasticity of cellular processes that drive tumorigenesis. Drug synthesis has been a cat and mouse game and it is likely, that this will continue at least for the foreseeable future. Nevertheless, this should not be interpreted as a failure to produce efficient treatments. On the contrary, our understanding of protein structure has in many cases allowed development of targeted small molecule drugs that are more efficient and selective than chemotherapy.

Understanding resistance mutations requires knowledge of organic chemistry and protein structure. In silico molecular modeling has also been helpful but often specialized software and computer programing knowledge is required. However, recently more user-friendly platforms are being developed. For example, COSMIC launched a platform known as COSMIC-3D to understand “cancer mutations in the context of 3D protein structure” (https://cancer.sanger.ac.uk/cosmic3d/) [607]. Interfaces such as this, will undoubtedly allow other scientists not necessarily trained as organic or protein biochemists to provide input and generate hypothesis that will be useful for drug development.

As seen in this review, many resistant mechanisms are similar. For example, when targeting kinases, certain “gatekeeper” residues are essential for facilitating drug–enzyme interaction. Mutations in these residues almost always affect the efficacy of the drug. Although each kinase is unique, patients have benefited from faster drug discovery when scientists compared different resistance mechanisms [608]. Comparisons of resistance mechanisms will accelerate next generation drug development.

Immunotherapy (briefly mentioned in this review) and modulation of the function of various non-coding RNAs are other forms of cancer therapies that we did not have time to address here [609,610,611,612,613,614]. The immune system function in slowing tumor progression has been well documented [615,616]. Non-coding RNAs have a plethora of functions in the cell including regulation of gene expression and protein localization [611]. Non-coding RNAs also regulate gene expression in the immune cells allowing tumors to evade the immune system [617]. One challenge has been the delivery of these therapies. However, with technical advances such as CRISPR [618,619] and nanotechnology [620,621,622,623,624] it is clear that in the near future we will see much more complex forms of cancer treatments.

## Figures and Tables

**Figure 1 cancers-12-00927-f001:**
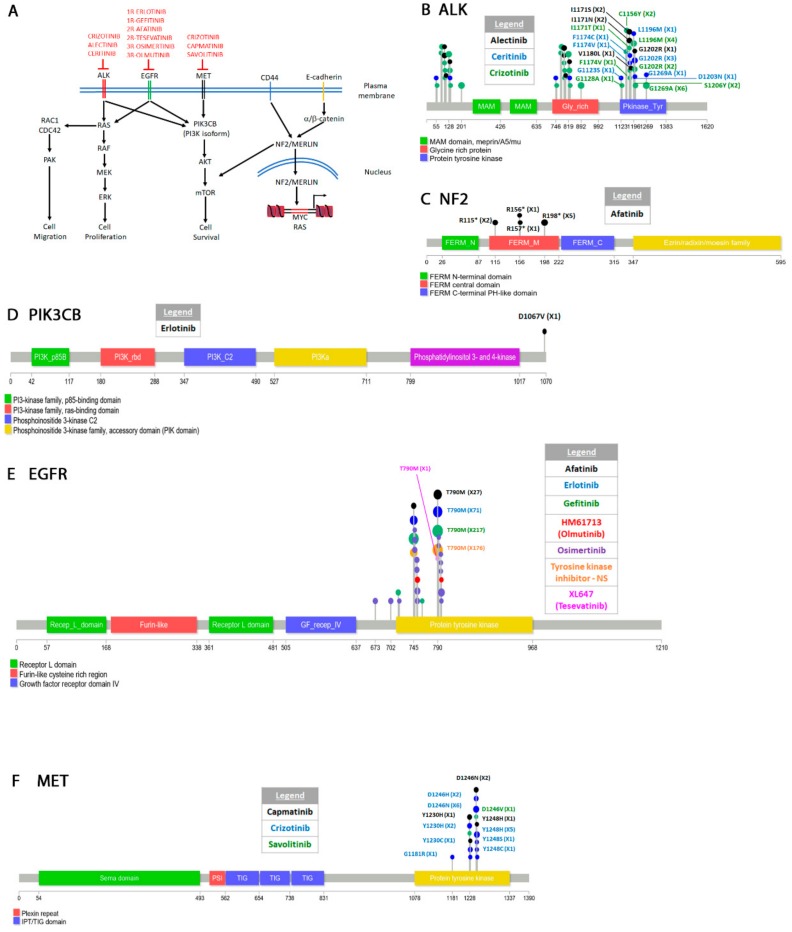
Signal transduction pathways in non-small cell lung cancers (NSCLC) and drug resistant mutations. (**A**) Some of the signaling pathways targeted by drugs that are discussed in this paper. Three receptor tyrosine kinases (ALK, EGFR, and MET) acquire resistant mutations to the drugs shown in red. For EGFR 1st, 2nd, and 3rd generation drugs are indicated as 1R, 2R, and 3R respectively. The CD44-cadherin-NF2/Merlin pathway that activates transcription of the MYC and RAS proto-oncogenes is also shown. Activating mutations in the NF2/merlin arise in response to drugs that target ALK, EGFR, and MET. (**B–F**) Diagrams showing the position of the mutations in several genes shown in (**A**). Note, that in order to not over-clutter the diagrams only some of the mutations are shown for certain genes. A comprehensive list is found in the Appendix A.

**Figure 2 cancers-12-00927-f002:**
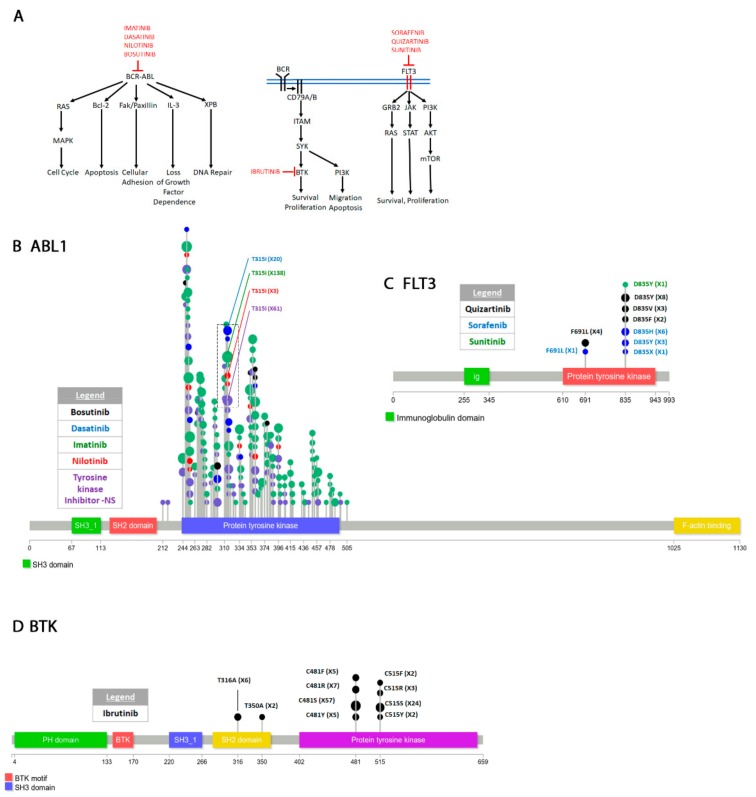
Pathways targeted in cancers of the hematopoietic and lymphoid tissue and drug resistant mutations. (**A**) The fused BCR-ABL protein signals through several pathways to activate cell cycle, apoptosis, adhesion, DNA repair and loss of growth factor dependence. The drugs that target the BCR-ABL fusion are shown. Some of these drugs also target the ABL1 wild type protein. Please see text for discussion. The pathways involving BTK and the FLT3 receptor tyrosine kinase are also shown because they are discussed in the text. (**B**) Mutations in the ABL1 gene cluster in the protein tyrosine domain. Only a few of the mutations are labeled to prevent clutter. Please see Appendix A for all mutations. (**C**,**D**) Resistant mutations in FLT3 and BTK.

**Figure 3 cancers-12-00927-f003:**
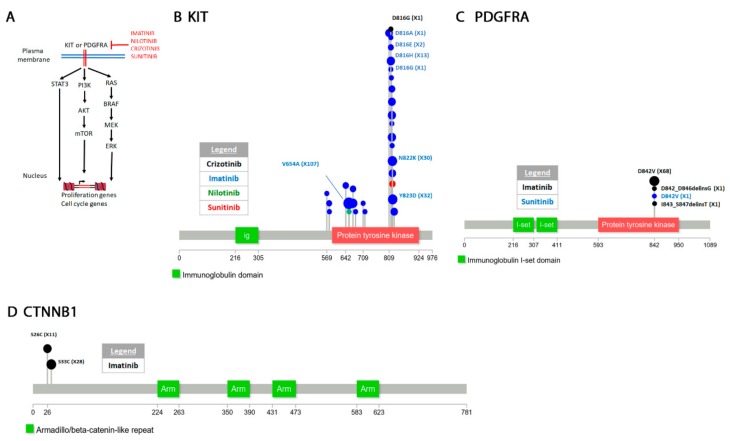
Signaling and mutations arising in gastrointestinal stromal tumors (GIST) soft tissue. (**A**) The KIT or PDGFRA receptor tyrosine kinases signal to at least three pathways to activate transcription of proliferation and cell cycle genes. The two receptor tyrosine kinases (RTKs) acquire resistance in response to the drugs shown. (**B–D**) Diagrams of resistant mutations in some of the genes targeted by the drugs shown in (**A**).

**Figure 4 cancers-12-00927-f004:**
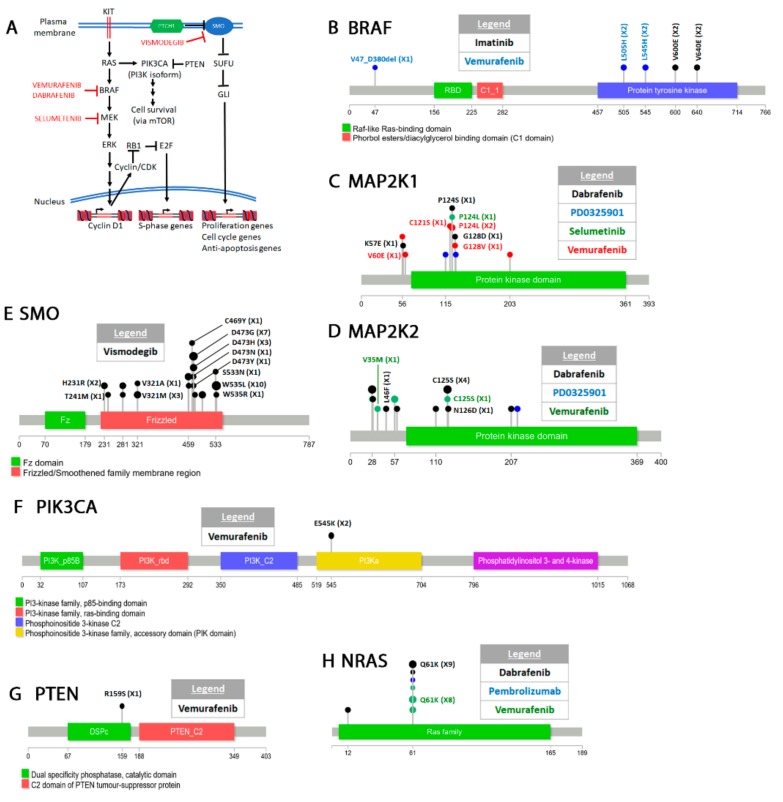
Drugs that target melanoma pathways. (**A**) Some of the signal transduction pathways discussed here and the drugs that target them. Please see text for details. (**B**) Drug resistant point mutations arising in BRAF. Also shown is one structural alteration. Other resistant structural alterations of BRAF are listed in Appendix A. (**C–F**) Secondary mutations acquired in the other genes involved in the pathways shown in (**A**).

**Figure 5 cancers-12-00927-f005:**
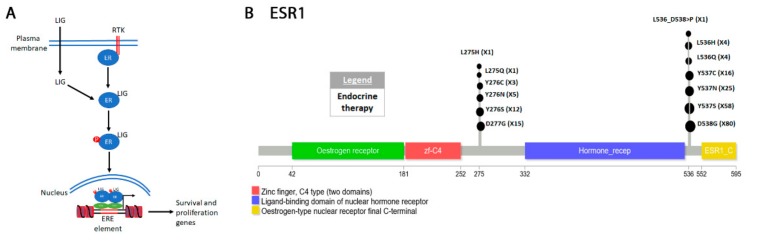
ERalpha (ESR1) in breast cancer. (**A**) Upon binding to its ligand the estrogen receptor translocates to the nucleus where it interacts with estrogen response elements to activate survival and proliferation genes. Estrogen receptor (ER) phosphorylation also appears to be required for activation. Not discussed here are the roles of the ER receptor in activating cytoplasmic signal transduction pathways through interaction with plasma membrane receptors [496]. (**B**) Resistant mutations in ESR1 arising in response to endocrine therapy.

**Figure 6 cancers-12-00927-f006:**
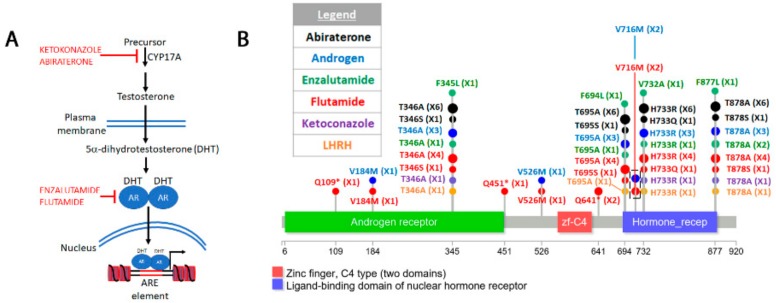
The androgen receptor (AR) signaling and drug resistant mutations. (**A**) Diagram of the signal transduction pathway for AR signaling. Testosterone is synthesized from a precursor in several epistatic biochemical steps. CYP17A is an enzyme involved in one of these steps. Testosterone crosses the plasma membrane of the target cell and is converted to 5α-dihydrotestosterone (DHT). DHT interacts with the AR receptor and the complex translocates to the nucleus where it binds androgen response elements (ARE) and activates gene transcription. The drugs that inhibit either CYP17A or AR are shown. (**B**) Diagram of resistant mutations in the androgen receptor. Only the amino acids discussed are shown.

**Table 1 cancers-12-00927-t001:** Summary of secondary mutations acquired by various genes in response to small molecule inhibitors.

Drug Name	Targeted Cancer Type(s)	Gene(s) Acquiring Resistant Mutation(s)	Primary Tissue where Resistant Mutations were Identified	Tumor Type where Resistant Mutations Identified ^1^
Erlotinib	**Non-Small Cell Lung Cancer (NSCLC) (with EGFR activating mutations)**^2^ Pancreatic cancer ^3^	EGFR (Epidermal Growth Factor Receptor), MET (Hepatocyte Growth Factor Receptor)	Lung	Adenocarcinoma, non-small cell carcinoma, bronchioloalveolar and acinar adenocarcinoma
Gefitinib	**NSCLC (with EGFR activating mutations)** Breast cancer as well as several other cancers	EGFR, MET	Lung	Adenocarcinoma, non-small cell, squamous cell and pleomorphic carcinoma, mixed adenosquamous carcinoma, bronchioloalveolar, acinar and micropapillary adenocarcinoma
Afatinib	**NSCLC (with EGFR activating mutations)** Breast cancer	EGFR, MET, NF2 (Merlin)	Lung	Adenocarcinoma, non-small cell carcinoma, bronchioloalveolar adenocarcinoma
Osimertinib	**NSCLC (with EGFR activating mutations and T790M resistant mutation)**	EGFR, MET	Lung	Adenocarcinoma, non-small cell carcinoma
Olmutinib (HM61713)	**NSCLC (with EGFR activating mutations and T790M resistant mutation)**	EGFR	Lung	Adenocarcinoma
Tesevatinib (XL647)	**NSCLC (with EGFR activating mutations and T790M resistant mutation)** Polycystic kidney disease Kidney cancer	EGFR	Lung	Adenocarcinoma
Capmatinib	**NSCLC (with MET activating mutations or MET amplification)**	MET	Lung	Adenocarcinoma
Alectinib	**NSCLC (ALK positive)**	ALK (Anaplastic Lymphoma Kinase)	Lung	Adenocarcinoma, non-small cell carcinoma
Crizotinib	**NSCLC (ALK, ROS1(Proto-oncogene tyrosine-protein kinase ROS) positive)** Other ALK positive cancers	ALK, KIT (Proto-oncogene receptor tyrosine kinase), MET	Lung, Soft Tissue	Adenocarcinoma, non-small cell carcinoma, mixed adenosquamous carcinoma, squamous cell carcinoma, NS
Ceritinib	**NSCLC (ALK positive with resistant mutations to crizotinib)**	ALK	Lung	Adenocarcinoma, non-small cell carcinoma
Savolitinib	**NSCLC** Renal and gastric cancers	MET	Lung	Adenocarcinoma
Imatinib	**Philadelphia chromosome positive leukemia** **KIT positive GIST (Gastrointestinal stromal tumors)** Skin tumors	ABL1(Abelson Murine Leukemia Viral Oncogene Homolog 1), BRAF (serine/threonine-protein kinase B-Raf), KIT, PDGFRA(Platelet Derived Growth Factor Receptor Alpha), CTNNB1 (Beta catenin) AC058822.1 ^4^	Hematopoietic and lymphoid, Soft Tissue, Skin	Chronic myeloid leukemia, acute lymphoblastic leukemia, blast phase chronic myeloid leukemia, acral lentiginous, epithelioid, spindle, spindle and epithelioid, NS
Dasatinib	**Philadelphia chromosome positive leukemia (with some imatinib resistant mutations)**	ABL1	Hematopoietic and lymphoid	Acute lymphoblastic leukemia, chronic myeloid leukemia, blast phase chronic myeloid leukemia
Nilotinib	**Philadelphia chromosome positive leukemia (with some imatinib resistant mutations)** GIST (KIT driven tumors)	ABL1, KIT	Hematopoietic and lymphoid, Soft Tissue	Chronic myeloid leukemia, blast phase chronic myeloid leukemia, spindle
Bosutinib	**Philadelphia chromosome positive leukemia (with some imatinib resistant mutations)**	ABL1	Hematopoietic and lymphoid	Chronic myeloid leukemia, blast phase chronic myeloid leukemia
Tyrosine Kinase Inhibitor-NS ^5^	NSCLC and leukemias	ABL1, EGFR	Hematopoietic and lymphoid, Lung	Adenocarcinoma, chronic myeloid leukemia, acute lymphoblastic leukemia, non-small cell carcinoma, blast phase chronic myeloid leukemia
Ibrutinib	**Lymphomas and Chronic lymphocytic leukemia** Other B-cell cancers	BTK (Bruton Tyrosine Kinase)	Hematopoietic and lymphoid	Chronic lymphocytic leukemia, lymphoplasmacytic lymphoma, mantle cell lymphoma
Quizartinib, Sorafenib	**Acute myeloid leukemia** Kidney and liver cancers	FLT3 (FMS-like Tyrosine Kinase 3)	Hematopoietic and lymphoid	Acute myeloid leukemia
Sunitinib	**GISTs (usually imatinib resistant)** AML, kidney cancer	FLT3, KIT, PDGFRA	Hematopoietic and lymphoid, Soft Tissue	Acute myeloid leukemia, NS
Vemurafenib	**Melanoma**	BRAF, MAP2K1/2(Mitogen activated protein kinase 1/2), NRAS (Neuroblastoma Ras viral oncogene homolog), PIK3CA(phosphatidylinositol-4,5-bisphosphate 3-kinase alpha), PTEN (Phosphatase and tensin homolog)	Skin, NS	Malignant melanoma, NS
Dabrafenib	**Melanoma** NSCLC with trametinib combination	BRAF, MAP2K1/2, NRAS	Skin, NS	Malignant melanoma
Vismodegib	**Basal cell carcinoma** **Gorlin syndrome** Small cell lung cancer, other cancers	SMO (Smoothened)	Skin, Central Nervous System, NS	Basal cell carcinoma, NS
Selumetinib	**Melanoma** NSCLC	MAP2K1	Skin	Malignant melanoma
Pembrolizumab	**Melanoma** NSCLC, other cancers	JAK1/2 (Janus kinase 1/2), NRAS	Skin, NS	Malignant melanoma
Endocrine Therapy	**Breast cancer**	ESR1 (Estrogen receptor alpha)	Breast	ER-positive carcinoma, ductal carcinoma, lobular carcinoma, ductolobular carcinoma
Rapamycin	**Breast cancer** Numerous other diseases Prevention of organ transplant rejection	MTOR (Mammalian target of rapamycin)	Breast	NS
PD0325901	**Breast, GISTs** Various other cancers	MAP2K1/2	Breast, Large Intestine	Adenocarcinoma, NS
Androgen Ablation	**Prostate cancer**	Androgen Receptor	Prostate	Adenocarcinoma, NS
Abiraterone	**Castration resistant prostate cancer**	Androgen Receptor	Prostate	Adenocarcinoma, NS
Ketoconazole, LHRH (Luteinizing hormone releasing hormone)	**Castration resistant prostate cancer**	Androgen Receptor	Prostate	NS
Enzalutamide	**Castration resistant prostate cancer**	Androgen Receptor	Prostate	Adenocarcinoma, NS
Flutamide	**Castration resistant prostate cancer**	Androgen Receptor	Prostate	NS
Infigratinib (BGJ398)	**Biliary tract cancer with sorafenib resistant mutations**	FGFR2 (Fibroblast growth factor receptor 2)	Biliary Tract	Cholangiocarcinoma
PF-04217903	**Kidney tumor**	MET	Kidney	Papillary renal cell carcinoma
Everolimus	**Thyroid cancer** Other cancers	MTOR	Thyroid	Anaplastic carcinoma

^1^ In some cases the tumor type is not specified and is listed as NS. ^2^ In **bold** are cancer types primarily discussed in this review. ^3^ Other cancer types and diseases for which the drug has been used or considered as a treatment option. ^4^ AC058822.1 is a fusion between FIP1L1 and PDGFRA associated with Hypereosinophilic syndromes [22]. ^5^ In some cases the drug is listed only as non-specified tyrosine kinase inhibitor.

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
