# Peer review of "Secondary Resistant Mutations to Small Molecule Inhibitors in Cancer Cells"

_cancers, 2020, doi:10.3390/cancers12040927_

Round 1

Reviewer 1 Report

In this review entitled "Secondary resistant mutations to small molecule inhibitors in cancer cells", Abdulaziz Bazi Hamid and Ruben Petreaca provided a comprehensive review of secondary resistant mutations that are reported in COSMIC for many types of cancers in hematopoietic as well as non-hematopoietic cancers.
I have few minor comments:
  1. There many spelling mistakes (especially in the names of some targeted therapies). I counted at least 10 and it needs to be reviewed carefully.
  2. Figures are very fuzzy which make it not readable.
  3. In Table 1, it will be more informative to add a column with a title: Targeted cancer type (Primary vs others). For example, Crizotinib and other ALK inhibitors are targeting ALCL (Anaplastic Large Cell Lymphoma).
  4. It would better to avoid some expressions such as "thanks to ..." or " beautifully outlines".

Author Response

We thank the reviewers for their careful reading of our manuscript. Below we provide point by point responses.  

Reviewer 1

In this review entitled "Secondary resistant mutations to small molecule inhibitors in cancer cells", Abdulaziz Bazi Hamid and Ruben Petreaca provided a comprehensive review of secondary resistant mutations that are reported in COSMIC for many types of cancers in hematopoietic as well as non-hematopoietic cancers.

I have few minor comments:

1.      There many spelling mistakes (especially in the names of some targeted therapies). I counted at least 10 and it needs to be reviewed carefully.

Thank you for your careful reading of the manuscript. We read it carefully to fix spelling and grammatical errors.

2.      Figures are very fuzzy which make it not readable.

We updated the figures in a higher resolution format.

3.      In Table 1, it will be more informative to add a column with a title: Targeted cancer type (Primary vs others). For example, Crizotinib and other ALK inhibitors are targeting ALCL (Anaplastic Large Cell Lymphoma).

We included another column in Table 1. However, we thought it would be more informative to list cancers that are specifically addressed in this review vs other cancers/diseases that the drugs are treating.

4.      It would better to avoid some expressions such as "thanks to ..." or " beautifully outlines".

We made sure to rephrase these sentences.

Reviewer 2 Report

This is an authoritative review of the role of acquired resistance and its impact on cancer therapeutics. The authors reviewed the Catalogue of Somatic Mutations in Cancer (COSMIC) to cull data on all resistance mutations for a number of kinases. Specifically, mutations in response to 35 drugs targeting 22 proteins were surveyed. The detailed literature survey provides a roadmap that enables the understanding of how these various mutations lead to common pathways of resistance.

Because the overall goal is to develop drugs, in this case, (tyrosine) kinase inhibitors, there shoud be a statement regarding how this information will be of benefit to patients with the various cancers discussed. This could be in the form of a hopeful statement in the Conclusions section.

Minor issues:

Page 1 line 11 (Abstract) abrogate

Page 2 line 64 as should be deleted

Page 3 bottom of Table 1 – should be “Everolimus”

Page 5 line 112 Legend to Fig. 1 – “in the several genes” should read “in several”

Page 6 line118 should read decreases

Page 6 line 123 should read “with EGFR activating mutations other than L858R”

Page 6 line 139 should read “after treatment with”

Page 6 line139 should read “mouse” not mice

Page 6 line 154 should read “cetuximab”

Page 7 line 207 should read “pocket”

Page 8 line 232 should read “xenograft”

Page 9 line 276 should read “conformational” and “increases”

Page 9 line 289 should read “resistance”

Page 11 line 367 should read “mutations”

Page 12 line439 should read “A team at”

Page 13 line 458  should read “cross-phosphorylates and activates the kinase”

Page 17 line 578  should read “binding”

Page 20 legend to Fig. 6 line 718  should read “translocates”

Page 21 line756 should read “the function of fibroblast”...

Page 21 line 780 the word “in” should be deleted.

Author Response

We thank the reviewers for their careful reading of our manuscript. Below we provide point by point responses.  

Reviewer 2

This is an authoritative review of the role of acquired resistance and its impact on cancer therapeutics. The authors reviewed the Catalogue of Somatic Mutations in Cancer (COSMIC) to cull data on all resistance mutations for a number of kinases. Specifically, mutations in response to 35 drugs targeting 22 proteins were surveyed. The detailed literature survey provides a roadmap that enables the understanding of how these various mutations lead to common pathways of resistance.

Because the overall goal is to develop drugs, in this case, (tyrosine) kinase inhibitors, there should be a statement regarding how this information will be of benefit to patients with the various cancers discussed. This could be in the form of a hopeful statement in the Conclusions section.

 We expanded the concluding remarks section to add this statement.

Minor issues:

Thank you for the careful reading of our manuscript. We made the indicated changes to every suggestion below.  

Page 1 line 11 (Abstract) abrogate

Page 2 line 64 as should be deleted

Page 3 bottom of Table 1 – should be “Everolimus”

Page 5 line 112 Legend to Fig. 1 – “in the several genes” should read “in several”

Page 6 line118 should read decreases

Page 6 line 123 should read “with EGFR activating mutations other than L858R”

Page 6 line 139 should read “after treatment with”

Page 6 line139 should read “mouse” not mice

Page 6 line 154 should read “cetuximab”

Page 7 line 207 should read “pocket”

Page 8 line 232 should read “xenograft”

Page 9 line 276 should read “conformational” and “increases”

Page 9 line 289 should read “resistance”

Page 11 line 367 should read “mutations”

Page 12 line439 should read “A team at”

Page 13 line 458  should read “cross-phosphorylates and activates the kinase”

Page 17 line 578  should read “binding”

Page 20 legend to Fig. 6 line 718  should read “translocates”

Page 21 line756 should read “the function of fibroblast”...

Page 21 line 780 the word “in” should be deleted.